# Sero-Epidemiology of *Pneumocystis* Infection among Infants, Children, and Adults in Chile

**DOI:** 10.3390/jof8020136

**Published:** 2022-01-29

**Authors:** Robert F. Miller, Kieran R. Daly, Peter D. Walzer, Ana V. Ulloa, Carolina A. Ponce, Sergio L. Vargas

**Affiliations:** 1Centre for Clinical Research in Infection and Sexual Health, Institute for Global Health, University College London, London WC1E 6JB, UK; robert.miller@ucl.ac.uk; 2Clinical Research Department, Faculty of Infectious and Tropical Diseases, London School of Hygiene and Tropical Medicine, London WC1E 7HT, UK; 3Division of Infectious Diseases, University of Cincinnati, Cincinnati, OH 45236, USA; kieran.daly@uc.edu (K.R.D.); peter.walzer1@gmail.com (P.D.W.); 4Instituto de Ciencias Biomédicas (ICBM), Facultad de Medicina Universidad de Chile, Santiago 8380453, Chile; lir@med.uchile.cl (A.V.U.); cponce@med.uchile.cl (C.A.P.)

**Keywords:** *Pneumocystis*, epidemiology, serology, major surface glycoprotein

## Abstract

Previous serologic surveys show >80% of infants in Chile have anti-*Pneumocystis* antibodies by 2 years of age, but the seroepidemiology of *Pneumocystis* infection beyond infancy is unknown. We describe the sero-epidemiology in infants, children, and adults at different locations in Chile. Serum samples were prospectively obtained from 681 healthy adults (age ≥ 17 years) and 690 non-immunocompromised infants/children attending eight blood banks or outpatient clinics (2 in Santiago) in Chile. ELISA was used to measure serum IgM and IgG antibodies to *Pneumocystis jirovecii* major surface antigen (Msg) constructs MsgA and MsgC1. Serologic responses to *Pneumocystis* Msg showed a high frequency of reactivity, inferring infection. Among infants/children increasing age and the proportion with detectable IgM responses to MsgA, and IgG responses to MsgA, and MsgC1 were positively associated. Among adults there was almost universal seropositivity to one or more *Pneumocystis* Msg constructs. In infants and children rates of detectable IgM responses to MsgC1 and MsgA were greater than IgG responses. In Santiago, rates of seropositivity among infants/children were greater in clinics located in a more socio-economically deprived part of the city. In Chile, a serological response to *Pneumocystis* Msg constructs was common across ages regardless of geographical location and climatic conditions. Observed higher rates of IgM responses than IgG responses is consistent with concept of recent/ongoing exposure to *Pneumocystis* in children and adults. Higher rates of seropositivity in infants/children residing in more densely populated areas of Santiago infers crowding poses an increased risk of transmission.

## 1. Introduction

*Pneumocystis* provokes a severe fungal disease known as *Pneumocystis* pneumonia (PcP) in immunocompromised patients in Central and South America [1,2,3,4,5,6,7,8,9], in low- and middle-income countries in Africa or Europe [10,11,12], and in the more industrialized world [13,14]. Serologic surveys for anti-*Pneumocystis* antibodies do not correlate with incidence of PcP. However, they have contributed to documenting the widespread occurrence of *Pneumocystis* in humans. They lead to a better understanding of the epidemiology and may reveal predisposing conditions and be useful to describe the efficacy of strategies of control or prevention.

Serologic surveys show that seroconversion resulting from the primary infection is common in children [15,16,17]. Furthermore, serologic surveys among children and adults in United States, United Kingdom, Uganda and Gambia show evidence of antibody response to infection in a majority of heathy and immunosuppressed populations [18,19,20]. Serologic surveys conducted in Chile show that more than 80% of infants have detectable anti-*Pneumocystis* antibody levels by 2 years of age, however, the seroepidemiology of *Pneumocystis* infection in Chile has not been studied beyond infancy [17,21].

The unique geography of the country permits the study of antibody levels in populations living at different elevations above sea level, with differing climatic conditions, and at latitudes that are south of Africa and Australia.

In this study, we report a high frequency of anti-*Pneumocystis* antibodies in healthy infants, children, and adults attending hospital outpatient clinics distributed along Chile, suggesting *Pneumocystis* is similarly prevalent and widely distributed at various geographical locations.

## 2. Methods

### 2.1. Ethics

The study was approved by the University of Chile School of Medicine Ethics Committee for Studies in Humans under approval letter 3446. One sample was required per participant. The study was conducted in accordance with the declaration of Helsinki. Written informed consent was obtained as indicated by local research laws and regulations. Except for age, samples were anonymized before analysis. The Institutional Review Board of the University of Cincinnati and the University of Chile School of Medicine Ethics Committee for Studies in Humans approved antibody analyses in Cincinnati, OH, USA.

### 2.2. Patients

Adult patients (age ≥ 17 years) and infants/children attending eight outpatient clinics in Chile, between January 1997 and April 1998 were studied. The clinics were located in Coquimbo, Andacollo, Canela, Santiago (two pediatric clinics and two blood banks at Hospital San José and Hospital Luis Calvo Mackenna), Temuco, Valdivia, and Paillaco (Figure 1). Serum samples were prospectively collected after signed informed consent and each individual provided one serum sample. Samples from adults were obtained from prospective blood donors, and the parents of healthy infants and children were approached at the blood sampling unit when they went for blood draws prior to surgical procedures or blood tests that were not related to immunosuppressive conditions. Individuals with any immunosuppressive conditions were excluded. None of the infants or adult participants had previously been diagnosed with, or were suspected of having, *Pneumocystis* pneumonia. Serum samples were stored at −70 °C until analyzed. With the exception of Canela, a very small rural community where only 9 children and 42 adults were recruited, an average of 97 samples from infants and children (range 70–106; median 100) and 91 samples from adults (range 78–100; median 95) per locality were obtained. The number of samples (approx. 100 per locality) was decided empirically and ultimately depended on availability. This was a first exploratory study of antibodies in different cities.

### 2.3. Recombinant Msg Constructs

The major surface glycoprotein (Msg) of *P. jirovecii* represents a family of proteins encoded by multiple genes. Recombinant *P. jirovecii* antigens such as msg provide a powerful tool for studying the interaction between the human host and *P. jirovecii* [22]. Three overlapping recombinant fragments that span the *P. jirovecii* Msg have been developed, MsgA (amino terminus), MsgB (middle) and MsgC1 (carboxyl terminus). The previously described recombinant MsgA and MsgC1 fragments [23] were prepared by PCR using DNA isolated from *P. jirovecii* infected lung or cloned Msg genes as templates and AmpliTaq enzyme (Applied Biosystems, Foster City, CA, USA) to generate Msg gene segments. The PCR products were cloned into the pET30 vector (Novagen, Madison, WI, USA) and the recombinant Msg proteins were expressed in *Escherichia coli* and purified as previously described [23].

### 2.4. ELISA

IgG: Using a previously described ELISA protocol developed in our lab [24], we measured serum IgG responses to two recombinant Msg constructs: MsgA and MsgC1. We tested subject sera and standard reference sera against the recombinant Msg constructs, using phosphate-buffered saline (PBS) without Msg as the negative control. We corrected the reactivity of each serum specimen to Msg by subtracting the reactivity of each serum specimen to PBS without Msg (mean optical density (OD) with Msg—mean OD without Msg) and quantified the results using methods described by Bishop and Kovacs [25]. Briefly, we prepared a standard serum with specificity for each Msg construct by mixing the sera from 4 to 6 specimens with high reactivity for the specific construct. We selected these specimens from banks of sera from blood donors and HIV-infected patients. The standard pool for each Msg construct was defined as having a value of 100 U in 100 μL of a 1:100 dilution. We used the same standard pools throughout the study. From the standard pool, we generated a standard curve for each Msg construct on each day the assay was used. We used this curve to calculate the units of reactivity to the Msg construct. We diluted test serum samples at 1:100 to fit the linear portion of the standard curves, and we then calculated units of reactivity for each serum specimen. A value of >1 was regarded as positive.

IgM: The IgM antibody reactivity to MsgA and MsgC1 was measured using the same ELISA method described above for IgG with the following modifications: The secondary antibody was affinity-purified, HRP-labeled goat anti-human IgM (μ-chain) (KPL Products, Gaithersberg, MD, USA) instead of goat anti-human IgG (H&L) (KPL Products) [24]. The specificity of the anti-IgM reagent had previously been shown to be specific for μ-chain antibodies and did not react with IgG antibodies [15,24]

### 2.5. Environmental and Climatological Factors

For each clinic location details of geographical co-ordinates (latitude and longitude) and elevation above sea level were obtained from https://www.maps.ie/coordinates.html (accessed on 4 December 2019), and details of average annual rainfall and average annual temperature were obtained from https://www.worldweatheronline.com (accessed on 3 December 2019).

### 2.6. Statistics

Antibody level determinations for IgM responses and IgG responses to MsgA, and MsgC1, were evaluated at each location and were compared using χ^2^ and the variation according to age was determined by χ^2^ for trend. Overall associations between geographic (latitude, longitude, and elevation above sea level) and climatologic (average annual rainfall, average annual mean temperature) conditions at each clinic, and detection of IgM and IgG Msg antibodies were explored using Spearman rank test. For all analyses, *p* < 0.05 was regarded as significant.

## 3. Results

In total, 1371 subjects participated in the study: 681 adults, aged between 17 and 88 years (median age 35 years: interquartile range 26–65.3) and 690 infants and children aged between one month and 16.9 years (median age 5 years: interquartile range 2.1–10.4) were studied.

Overall, there was evidence of a high frequency of asymptomatic infection, as demonstrated by serologic responses to *Pneumocystis* Msg constructs. Among both infants/children and in adults, the frequency of IgG MsgA responses were greater than MsgC1 responses (χ^2^ = 208.9; *p* < 0.001, and χ^2^ = 102; *p* < 0.001, respectively), whereas IgM responses to MsgA and MsgC1 were similar (χ^2^ = 0.176; *p* = 0.675), and the frequency of IgM MsgC1 responses were slightly greater than MsgA responses in children/infants (χ^2^ = 7.432; *p* = 0.006) (Table 1 and Table 2). In infants and children, the highest frequency of antibody responses was seen in Andacollo (IgM responses to MsgC1 and MsgA) and in Temuco and Valdivia (IgM responses to MsgC1) and in Hospital San José in Santiago (IgG responses to MsgC1 and MsgA) (Table 1). There were strikingly lower median values and rates of serological responses in samples obtained at the Hospital Luis Calvo Mackenna when compared to samples from the Hospital San José: IgG MsgA = 88.8% vs. 99% (χ^2^ = 9.336; *p* = 0.002) and IgG Msg C1 60% vs. 78.8% (χ^2^ = 9.336; *p* = 0.002) (Table 1).

Among infants and children there was evidence of an association between increasing age and the proportion with detectable IgM responses to MsgA, IgG responses to MsgA, and IgG responses to MsgC1 antibodies: χ^2^ for trend = 11.57; *p* = 0.0007, 22.64; *p* < 0.0001, and 36.5; *p* = 0.001, respectively. There was no evidence of an association between increasing age and detection of IgM MsgC1 antibodies: χ^2^ for trend = 3.731; *p* = 0.0534. Among adults there was no evidence of an association between increasing age and the proportion with detectable IgM responses to MsgA, IgM responses to MsgC1, IgG responses to MsgA and IgG responses to MsgC1; χ^2^ for trend = 0.0486; *p* = 0.8255, 0.874: *p* = 0.3499, 0.3682; *p* = 0.5440, and 0.0122; *p* = 0.9119, respectively.

Among adults there was almost universal seropositivity to one or more *Pneumocystis* Msg constructs with the highest median of detectable serological responses observed in Andacollo (IgG responses to Msg A) and Canela (IgG responses to MsgA, IgM responses to MsgA, and Msg C1). All adults in Hospital Luis Calvo Mackenna, in Santiago were seropositive for *Pneumocystis* IgM (Table 2).

There was no evidence of an association between the mean annual temperature at each clinic site and the frequency of detection of IgM responses to MsgA, IgM responses to MsgC1, and IgG responses to MsgA or MsgC1 (Table 1, Table 2 and Table 3). Additionally, there was no evidence of an association between average annual rainfall, or between geographical co-ordinates (latitude and longitude), and elevation above sea level, and frequency of detection of *Pneumocystis* Msg antibodies.

## 4. Discussion

This study describes the sero-epidemiology of *Pneumocystis* infection among infant, children and adult populations in different cities in Chile. It demonstrates that a serological response to *Pneumocystis* is present across ages regardless of geographical location and suggests that infection, as measured by Msg antibody responses may be more prevalent in areas of Santiago with a poorer a socio-economic situation. Overcrowding conditions have been recently shown to be associated to increased colonization by *Pneumocystis* [26].

Across all centers variations in rates of seropositivity with increasing age were observed among infants and children, but not among adults, and despite large differences in geographical factors (latitude, longitude, and elevation above sea level), and average rainfall at each center no association with rates of seropositivity was demonstrated. No association between average annual temperature and seropositivity was seen, however there is little variation in mean average temperature between the study locations.

In Santiago infants/children from San José Hospital had a higher median and rate of seropositivity than those providing samples at the Calvo Mackenna Children’s Hospital. The difference may be attributable to the subjects’ differing socio-economic conditions, including poverty, and over-crowded housing, as these two hospitals serve different populations within the geographical area of the city. Calvo Mackenna Hospital is a general pediatric hospital that hosts the national referral center for bone marrow and liver transplants, and a large pediatric oncology unit, whereas Hospital San José serves a widespread catchment area of the general population, with a wider differential diagnosis on presentation to the clinic, which is located in the North of the metropolis, an area with less advantageous socio-economic conditions. As increasing age among infants and children was associated with the rate of detection of Msg antibodies, age differences between subjects at the two hospitals might have explained this difference, if children were older at Hospital San José, as they would have had a greater lifetime risk of exposure/acquisition of infection. However, children at Hospital Calvo Mackenna were aged 0.2–14.3 years (median 5.4), and at Hospital San José they were aged 0.2–16.6 years (median 5.5).

An explanation for the frequency of different antibody responses to MsgA compared to MsgC1 might be due to selection of antigenic epitopes. Selection of antigenic epitopes during an immune response is due in part to the Major Histocompatibility Complex molecules and in part to the B and T cell repertoires of individuals. We do not know the spectrum of Msg molecules that may have been encountered during exposure to *Pneumocystis* in our study population and thus, may have been the original antigenic stimulus. It is possible that there are more antigenic epitopes contained within MsgA than MsgC1. Should more epitopes exist on a given antigen, the probability of finding reactive antibodies may increase compared to an antigen with fewer epitopes. Another possibility is that MsgC1 sequences may be more variable than MsgA sequences. This infers that MsgA portions may be similar to one another, have more cross-reactive epitopes and, therefore, use of MsgA used in ELISA detects more positive responses. On the other hand, MsgC1 is highly variable, having fewer cross-reactive epitopes and therefore, the choice of MsgC1 used in ELISA will likely yield fewer positive responses. Additionally, we were not able to test other variants of MsgA or MsgC1 in our study due to limitations in sample serum volume.

Primary *Pneumocystis* infection is probably acquired by infants from their colonized mothers, either trans-placentally, or soon after birth via the airborne route [21,27,28,29,30,31]. A search for *Pneumocystis* using molecular diagnostics has additionally permitted detection of a peak of the primary infection by *Pneumocystis* between 2 and 5 months of age and subclinical infections in healthy individuals that imply wide circulation of *Pneumocystis* organisms in the community [32,33,34]. This study shows that in infants and older children, rates of detectable IgM responses to MsgC1 and MsgA were greater than IgG responses. This observation is consistent with the concept of repeated/recent/ongoing exposure to *Pneumocystis* throughout childhood, given that IgM antibody responses to *P. jirovecii* infer acute/recent exposure.

*Pneumocystis* has been reported in all continents as far north as Trondheim, Norway, (latitude = 63°26′) but not north of the Arctic circle (latitude 66°33′ N) or in Antarctica (south of latitude 66°33′). Detection of a serological response to *Pneumocystis* in Chile as far south as Paillaco (latitude 40°0′ S), which is further south than Cape Town, South Africa (33°9′ S), and Sydney, Australia (33°8′ S), confirms the widespread distribution of the infection, and adds to reports of *Pneumocystis* pneumonia at extremely southern latitudes including Hobart, Tasmania [35] (latitude 42°8′ S), and Christchurch, New Zealand [36] (latitude 43°5′ S).

This study has some limitations. First, the number of study participants in some cities was relatively small, and no information about housing conditions was obtained. Second, only a small number of study sites were included. Third, clinics in very hot and dry environments (e.g., Iquique, Calama; latitude 20°21′ S) and those in cold and wet locations (e.g., Coyhaique, Punta Arenas; latitude 45°34′ S) were not included in this first sampling of cities. Description of the sero-epidemiology of *Pneumocystis* infection at such extreme latitudes or climatic conditions needs to be studied, as it would add to “the body of knowledge”. Fourth, the effect of the season of the year was not considered. Thus, the potential impact of environmental factors on acquisition of *Pneumocystis* infection deserves to be explored further.

## 5. Conclusions

This study adds to the body of knowledge about the epidemiology of *Pneumocystis* infection. It demonstrates that *Pneumocystis* is frequent across ages in many areas of Chile and suggests that in a city environment, overcrowding associates with increased rates of antibody detection among infants and children. Future research should extend sampling into other populations, for example, adult healthy blood donors living in more extreme environmental conditions, as well as exploring the potential for seasonal variation. Additionally, socio-economic factors that associate with poverty and rates of *Pneumocystis* infection should be further explored.

## Figures and Tables

**Figure 1 jof-08-00136-f001:**
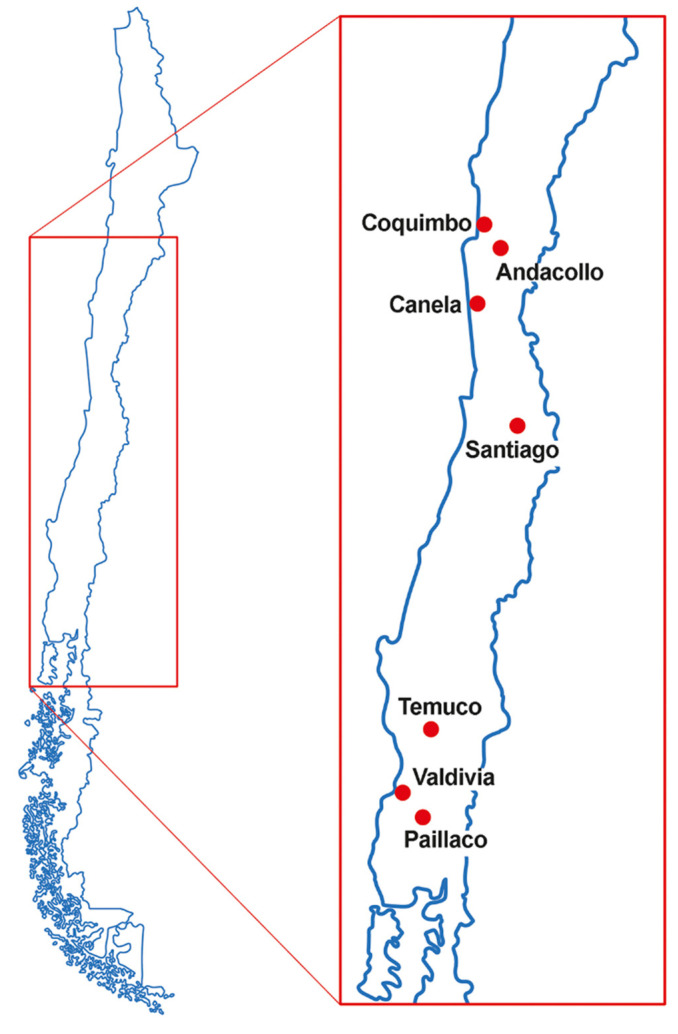
Map of Chile showing the location of the clinics.

**Table 1 jof-08-00136-t001:** Ig M and Ig G antibody responses to *Pneumocystis* MsgA and MsgC1 constructs in infants and children.

Clinic Location (Number of Subjects)	Ig M MsgA		Ig M MsgC1		Ig G MsgA		Ig G MsgC1	
	Number + (%)	Levels (Median:IQR)	Number + (%)	Levels (Median:IQR)	Number + (%)	Levels (Median:IQR)	Number + (%)	Levels (Median:IQR)
Coquimbo (N = 77)	70 (90.9)	23:10–50	66 (85.7)	28:13–52	68 (88.3)	23:11–36	38 (49.4)	23:15–44
Andacollo (N = 100)	99 (99)	31:19–50	99 (99)	34:21–51	88 (88)	20:9–33	60 (60)	14:7–28
Canela (N = 9)	9 (100)	21:11–34	9 (100)	19:9–27	9 (100)	21:9–64	6 (66)	13:11–36
Santiago Luis Calvo Mackenna (N = 98)	83 (84.7)	14:7–21	93 (94.9)	21:13–34	87 (88.7)	20:11–40	59 (60.2)	19:9–28
Santiago San José (N = 104)	93 (89.4)	15:9–26	99 (95.2)	23:13–44	103 (99)	47:24–81	82 (78.8)	37:24–51
Temuco (N = 94)	94 (100)	29:17–49	94 (100)	34:22–63	91 (97.8)	32:15–54	48 (51)	18:10–34
Valdivia (N = 109)	106 (97.2)	21:14–36	108 (99.1)	33:20–44	102 (93.6)	26:13–55	82 (75.2)	30:18–40
Paillaco (N = 101)	91 (90.1)	29:15–52	96 (95)	28:19–48	96 (95)	38:27–62	78 (77.2)	32:18–42

Key: + = positive. (%) = frequency of detection.

**Table 2 jof-08-00136-t002:** Ig M and Ig G antibody responses to *Pneumocystis* MsgA and MsgC1 constructs in adults.

Clinic Location (Number of Subjects)	Ig M MsgA		Ig M MsgC1		Ig G MsgA		Ig G MsgC1	
	Number + (%)	Levels (Median:IQR)	Number + (%)	Levels (Median:IQR)	Number + (%)	Levels (Median:IQR)	Number + (%)	Levels (Median:IQR)
Coquimbo (N = 81)	75 (92.5)	18:11–38	77 (95)	26:16–53	81 (100)	39:22–84	74 (91.4)	32:21–44
Andacollo (N = 98)	91 (93)	20:8–31	91(93)	28:8–49	97 (99)	49:20–74	77 (78.5)	27:13–38
Canela (N = 42)	42 (100)	39:15–60	42 (100)	35:15–60	41 (97.5)	53:32–91	35 (83)	29:17–40
Santiago Luis Calvo Mackenna (N = 100)	100 (100)	17:12–27	97 (97)	23:13–33	99 (99)	29:17–48	76 (76)	25:15–39
Santiago San José (N = 95)	92 (95.8)	26:14–57	92 (95.8)	29:16–50	94 (97.9)	38:17–66	78 (81.2)	26:17–41
Temuco (N = 78)	75 (96)	24:14–31	78 (100)	24:15–34	77 (98.7)	30:15–65	62 (79.5)	27:16–33
Valdivia (N = 89)	89 (100)	28:16–49	86 (96.6)	29:16–52	88 (98.9)	38:21–79	81 (91)	31:19–48
Paillaco (N = 98)	95 (97)	20:12–39	93 (94.9)	22:10–40	97 (99)	37:23–66	84 (85.7)	38:22–53

Key: + = positive. (%) = frequency of detection.

**Table 3 jof-08-00136-t003:** Geographic, environmental and climatologic conditions for each of the study locations in Chile.

Clinic Location	Geographical Location		Environmental and Climatologic Conditions				
	Latitude (Degrees)	Longitude (Degrees)	Elevation above Sea Level (m)	Average Annual Rainfall (mm)	Average Annual Temperature (°C)		
					Maximum	Minimum	Mean
Coquimbo	29°9′ S	71°3′ W	16	85	26.7	10.8	19.1
Andacollo	30°2′ S	71°0′ W	1019	97	19.6	15.2	18
Canela	31°4′ S	71°5′ W	200	169.5	21.1	6.7	14.2
Santiago	33°4′ S	70°6′ W	574	312	22.2	14.9	19
Temuco	38°7′ S	72°5′ W	120	1258	18.4	9.3	15.1
Valdivia	39°8′ S	73°2′ W	17	1752	16.8	9.6	13.4
Paillaco	40°0′ S	72°8′ W	93	1817	16	8.5	13

Geographical coordinates (latitude and longitude) and elevation above sea level for each clinic location were obtained from: https://www.maps.ie/coordinates.html [accessed on 8 March 2021]; average annual rainfall and temperature data were obtained from: https://www.worldweatheronline.com [accessed on 8 March 2021].

## Data Availability

Data is included in Table 1 and Table 2 and additionally available upon request.

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
