# Peer review of "Sero-Epidemiology of *Pneumocystis* Infection among Infants, Children, and Adults in Chile"

_jof, 2022, doi:10.3390/jof8020136_

Round 1
Reviewer 1 Report
The authors present epidermiological data in infants children and adults with seropositivity for Pneumocystis infections. The authors demonstrate that in Chile seropositivity to Pneumocytis infection is frequent across all ages and may be related to overcrowding. The manuscript adds to the data currently available on Pneumocystis infections.
Specific comments
1. Could the author postulate what effect on the results would have been if they had included areas with extreme climatic conditions which were not included as mentioned under limitations
2. Again the authors specified their numbers were relatively low using the data available in children could the authors estimate the calculate the minimum number of patients needed for this study-.
Author Response
RESPONSE TO REVIEWER 1 Comments and Suggestions for Authors
The authors present epidemiological data in infants children and adults with seropositivity for Pneumocystis infections. The authors demonstrate that in Chile seropositivity to Pneumocytis infection is frequent across all ages and may be related to overcrowding. The manuscript adds to the data currently available on Pneumocystis infections.
Specific comments
- Could the author postulate what effect on the results would have been if they had included areas with extreme climatic conditions which were not included as mentioned under limitations
RESPONSE
We thank the Reviewer for this observation and agree. We have modified our discussion to read:
“Third, clinics in very hot and dry environments (e.g. Iquique, Calama; latitude 20° 21’S) and those in cold and wet locations (e.g. Coyhaique, Punta Arenas; latitude 45°34’S) were not included in this first sampling of cities. Description of the sero-epidemiology of Pneumocystis infection at such extreme latitudes or climatic conditions needs to be studied as would add to “the body of knowledge”. Fourth, the effect of season of the year was not considered. Thus, the potential impact of environmental factors on acquisition of Pneumocystis infection deserves to be explored further. “
Again the authors specified their numbers were relatively low using the data available in children could the authors estimate the calculate the minimum number of patients needed for this study-.
RESPONSE
We are grateful to the Reviewer for asking this question. As this study was exploratory, we empirically aimed to obtain close to 100 samples per age per city depending on availability and were unable to calculate a minimum number. Given the high frequency of detection of both IgM and IgG antibody responses to Msg constructs at all sites in our study we suggest that the numbers studied are sufficient to support our observations. We have specifically suggested that our study might have been enhanced, had we included infants/children (and adults) from additional centres, particularly centres in the hot/dry and cold/wet parts of the country.
This sentence was added to the Methods section: “The number of samples obtained per city was decided empirically depending on availability as this was a first exploratory study of antibodies in different cities.”
Reviewer 2 Report
This papers reports search of antibodies against Pneumocystis jirovecii in a relatevely large cohort of immunocompetent children and adults from different cities in Chile. This study is original and of high interest in the field. However, it presently suffers from unperfect analyses and improvable presentation.
Major comments
- The nature of the two fragments MsgA and MsgC1 should be described and discussed : could the position of C1 closer to the cell membrane explain that less IgG are directed against it ?
- The paragraph lines 124 to 133 is unclear, presentation must be improved in order to be understanble to the reader.
- The paragraph lines 124 to 133 misses many statistical anylses so that the conclusions drawn in abstract and conclusions sections are not convincing. For example, lines 125-126, the following difference should be supported by statistical significance : " Among both infants/children and in 126 adults the frequency of IgG MsgA responses were greater than MsgC1 responses". All observations in this paragraph should be supported by statistical significance
- Line 148-149: unclear if association means higher frequency of detection of levels of Ig in blood associated with higher mean temperature. Does "detection" mean "frequency". The actual numbers that have been compared should be clarified
- Discussion and conclusions: the conclusions drawn should be supported by statistical significance.
- Line 202-203: the following sentence should be developed to increase clarity : "This observation is consistent with the concept of repeated/ recent/ongoing exposure to Pneumocystis throughout childhood."
Minor comments
- Line 17 abstract: it should stated that all children were immunocompetent, including those from outpatient clinics
- Line 20-22: the following sentence cannot be understood, is "antibodies" to be deleted ? : "Among infants/children increasing age and the proportion with detectable IgM responses to MsgA, and IgG responses to MsgA, and MsgC1 antibodies were positively associated."
- Line 84: reference 22 does appear to be correct one for these two Msg fragments.
- Line 120: delete "overall"
- Line 127: delete "adults" ? this sentence does not make sense, there could be copy/paste errors.
- Line 144: "serological responses" means frequency or levels or both ? in general, use of the same terms/words both in text and in the tables would improve the manuscript
- Line 146-147: also in Canela all adults are seropositive, which is not stated in the text. In general, the description of the results should be more rigourous
- Line 149 : add "tables 1a and 1b" because these tables contain the frequencies of detection that are analysis.
Author Response
RESPONSE TO REVIEWER 2 Comments and Suggestions for Authors
This paper reports search of antibodies against Pneumocystis jirovecii in a relatevely large cohort of immunocompetent children and adults from different cities in Chile. This study is original and of high interest in the field. However, it presently suffers from unperfect analyses and improvable presentation.
Major comments
- The nature of the two fragments MsgA and MsgC1 should be described and discussed : could the position of C1 closer to the cell membrane explain that less IgG are directed against it ?
RESPONSE
We thank Reviewer 2 for this suggestion regarding a better description of the two fragments MsgA and MsgC1.
We have amended the text of the methods to add text which reads:
“The major surface glycoprotein (Msg) of P. jiroveci represents a family of proteins encoded by multiple genes. Recombinant P. jirovecii antigens such as msg provide a powerful tool for studying the interaction between the human host and P. jirovecii. Three overlapping recombinant fragments that span the P. jirovecii Msg have been developed, MsgA (amino terminus), MsgB (middle) and MsgC1 (carboxyl terminus)22. The previously described recombinant MsgA and MsgC1 fragments…….”
Regarding the Reviewer’s suggestion that the observed differences may be due to the closer proximity of MsgC1 to the cell membrane - we have introduced a paragraph into the Discussion, which specifically addresses this issue:
“An explanation for the frequency of different antibody responses to MsgA compared to MsgC1 might be due to selection of antigenic epitopes. Selection of antigenic epitopes during an immune response is due in part to the Major Histocompatibility Complex molecules and in part to the B and T cell repertoires of individuals. We do not know the spectrum of Msg molecules that may have been encountered during exposure to Pneumocystis in our study population and thus, may have been the original antigenic stimulus. It is possible that there are more antigenic epitopes contained within MsgA than MsgC1. Should more epitopes exist on a given antigen, the probability of finding reactive antibodies may increase compared to an antigen with fewer epitopes. Another possibility is that MsgC1 sequences may be more variable than MsgA sequences. This infers that MsgA portions may be similar to one another, have more cross-reactive epitopes and therefore, use of MsgA used in ELISA detects more positive responses. On the other hand, MsgC1 is highly variable, having fewer cross-reactive epitopes and therefore, the choice of MsgC1 used in ELISA will likely yield fewer positive responses. Additionally, we were not able to test other variants of MsgA or MsgC1 in our study due to limitations in sample serum volume.”
- The paragraph lines 124 to 133 is unclear, presentation must be improved in order to be understandable to the reader.
RESPONSE
We thank the Reviewer for highlighting this issue and are we are grateful for the opportunity to improve this section.
Please see response to point 3 (below).
- The paragraph lines 124 to 133 misses many statistical analyses so that the conclusions drawn in abstract and conclusions sections are not convincing. For example, lines 125-126, the following difference should be supported by statistical significance : " Among both infants/children and in 126 adults the frequency of IgG MsgA responses were greater than MsgC1 responses". All observations in this paragraph should be supported by statistical significance
RESPONSE
We have re-written this paragraph, to include the Reviewer’s helpful suggestion regarding inclusion of statistical analyses in this section. In addition, the statistic analyses were re-done. The text now reads:
“Results. In total 1371 subjects participated in the study: 681 adults, aged between 17 and 88 years (median age 35 years: interquartile range 26-65.3) and 690 infants and children aged between one month and 16.9 years (median age 5 years: interquartile range 2.1-10.4) were studied.
Overall, there was evidence of a high frequency of infection, as demonstrated by serologic responses to Pneumocystis Msg constructs. Among both infants/children and in adults the frequency of IgG MsgA responses were greater than MsgC1 responses (c2 =208.9; p <0.001, and c2 =102; p <0.001, respectively), whereas IgM responses to MsgA and MsgC1 were similar in adults (c2 =0.176; p =0.675), and the frequency of IgM MsgC1 responses were greater than MsgA responses in children/infants (c2 =7.432; p =0.006) (Table 1a, Table 1b). In infants and children, the highest mean antibody responses were seen in Andacollo (IgM responses to MsgC1 and MsgA) and in Temuco-Valdivia (IgM responses to MsgC1) and in Hospital San José in Santiago (IgG responses to MsgC1 and MsgA) (Table 1a). There were strikingly lower median values and rates of serological responses in samples obtained at the Hospital Luis Calvo Mackenna when compared to samples from the Hospital San José: IgG MsgA =88.8% vs 99% (c2 =9.336; p =0.002) and IgG Msg C1 60% vs 78.8% (c2 =9.336; p =0.002) (Table 1a).”
- Line 148-149: unclear if association means higher frequency of detection of levels of Ig in blood associated with higher mean temperature. Does "detection" mean "frequency". The actual numbers that have been compared should be clarified
RESPONSE
We are grateful to have the opportunity to revise this point. We have re-done statistical analyses from the original data and detected typing mistakes that were corrected. The text was modified the text and now states:
“There was no evidence of an association between the mean annual temperature at each clinic site and the frequency of detection of IgM responses to MsgA, IgM responses to MsgC1, and IgG responses to MsgA or MsgC1 (Tables 1a, 1b, 2). Additionally, there was no evidence of an association between average annual rainfall, or between geographical co-ordinates (latitude and longitude), and elevation above sea level, and frequency of detection of Pneumocystis Msg antibodies.”
The actual numbers that have been compared are displayed in Tables 1a, 1b and Table 2).
- Discussion and conclusions: the conclusions drawn should be supported by statistical significance.
RESPONSE
We agree with Reviewer 2. Our Discussion and Conclusions have been enhanced by the inclusion of statistical analyses and p. values. Please see our responses to points 2, 3 and 4 (above).
- Line 202-203: the following sentence should be developed to increase clarity : "This observation is consistent with the concept of repeated/ recent/ongoing exposure to Pneumocystis throughout childhood."
RESPONSE
Thank-you for asking us to provide clarity on this point.
We have modified the text of the Discussion to state:
“This study shows that in infants and older children, rates of detectable IgM responses to MsgC1 and MsgA were greater than IgG responses. This observation is consistent with the concept of repeated/recent/ongoing exposure to Pneumocystis throughout childhood, given that IgM antibody responses to P. jirovecii infer acute/recent exposure.”
Minor comments
- Line 17 abstract: it should stated that all children were immunocompetent, including those from outpatient clinics
RESPONSE
Thank-you. We agree. The presence of any immunocompromising condition was exclusionary. Done.
- Line 20-22: the following sentence cannot be understood, is "antibodies" to be deleted ? : "Among infants/children increasing age and the proportion with detectable IgM responses to MsgA, and IgG responses to MsgA, and MsgC1 antibodies were positively associated."
RESPONSE
Thank-you for spotting this “typo”. The word “antibodies” was deleted.
- Line 84: reference 22 does appear to be correct one for these two Msg fragments.
RESPONSE
We thank the Reviewer for spotting this mistake. Reference [22] was misplaced and this was corrected.
- Line 120: delete "overall"
RESPONSE
Thank-you. “Overall” deleted as suggested.
- Line 127: delete "adults" ? this sentence does not make sense, there could be copy/paste errors.
RESPONSE
We thank Reviewer 2 for spotting this mistake. The word “adults” was deleted.
- Line 144: "serological responses" means frequency or levels or both? in general, use of the same terms/words both in text and in the tables would improve the manuscript
RESPONSE
Thank-you for asking us to clarify this point. We are referring to frequency of detection, and not to the level of detectable antibody response to any Msg Construct.
We have clarified this in the manuscript:
“Among adults there was almost universal seropositivity to one or more Pneumocystis Msg constructs with the highest rate of detectable serological responses observed in Andacollo……”
“(%)= frequency of detection” was also added below each table for clarification
- Line 146-147: also in Canela all adults are seropositive, which is not stated in the text. In general, the description of the results should be more rigorous
RESPONSE
We thank the Reviewer for asking for this additional information. We have enhanced our description of the results to show:
“Results
In total 1371 subjects participated in the study: 681 adults, aged between 17 and 88 years (median age 35 years: interquartile range 26-65.3) and 690 infants and children aged between one month and 16.9 years (median age 5 years: interquartile range 2.1-10.4) were studied.
Overall, there was evidence of a high frequency of infection, as demonstrated by serologic responses to Pneumocystis Msg constructs. Among both infants/children and in adults the frequency of IgG MsgA responses were greater than that of the MsgC1 responses (c2 =208.9; p <0.001, and c2 =102; p <0.001, respectively), whereas IgM responses to MsgA and MsgC1 were similar in adults (c2 =0.176; p =0.675), and the frequency of IgM MsgC1 responses were greater than that of MsgA responses in children/infants (c2 =7.432; p =0.006) (Table 1a, Table 1b). In infants and children, the highest frequency of antibody responses was seen in Andacollo (IgM responses to MsgC1 and MsgA) and in Valdivia (IgM responses to MsgC1) and in Hospital San José in Santiago (IgG responses to MsgC1 and MsgA) (Table 1a). There were strikingly lower rates of serological responses in samples obtained at the Hospital Luis Calvo Mackenna when compared to samples from the Hospital San José: IgG MsgA =88.8% vs 99% (c2 =9.336; p =0.002) and IgG Msg C1 60% vs 78.8% (c2 =9.336; p =0.002) (Table 1a).
Among infants and children there was evidence of an association between increasing age and the proportion with detectable IgM responses to MsgA, IgG responses to MsgA , and IgG responses to MsgC1 antibodies: c2 for trend =11.57; p =0.0007, 22.64; p <0.0001, and 36.5; p =0.001, respectively. There was no evidence of an association between increasing age and detection of IgM MsgC1 antibodies: c2 for trend =3.731; p =0.0534. Among adults there was no evidence of an association between increasing age and the proportion with detectable IgM responses to MsgA, IgM responses to MsgC1, IgG responses to MsgA and IgG responses to MsgC1; c2 for trend =0.0486; p =0.8255, 0.874: p =0.3499, 0.3682; p =0.5440, and 0.0122; p =0.9119, respectively.
Among adults there was almost universal seropositivity to one or more Pneumocystis Msg constructs with the highest rate of detectable serological responses observed in Andacollo (IgG responses to Msg A) and Canela (IgG responses to MsgA, IgM responses to MsgA, and Msg C1). All adults in Hospital Luis Calvo Mackenna, in Santiago were seropositive for Pneumocystis IgM (Table 1b).
There was no evidence of an association between the mean annual temperature at each clinic site and the frequency of detection of IgM responses to MsgA, IgM responses to MsgC1, and IgG responses to MsgA or MsgC1 (Tables 1a, 1b, 2). Additionally, there was no evidence of an association between average annual rainfall, or between geographical co-ordinates (latitude and longitude), and elevation above sea level, and frequency of detection of Pneumocystis Msg antibodies.
We hope this is a clearer, and more understandable description of our data.
- Line 149 : add "tables 1a and 1b" because these tables contain the frequencies of detection that are analysis.
RESPONSE
Thank-you. We agree. Done.

Round 2
Reviewer 2 Report
Thank you, the authors have responded very adequately to my comments.
Author Response
Response to Academic Editor comments:
Point 1: Please add a specific paragraph on the specificity of the antibodies used in this ELISA.
Response 1: The reactivity of IgG was already clearly stated and therefore we have included a paragraph on IgM reactivity as follows:
IgM: The IgM antibody reactivity to MsgA and MsgC1 was measured using the same ELISA method described above for IgG with the following modifications: The secondary antibody was affinity-purified, HRP-labeled goat anti-human IgM (m-chain) (KPL Products, Gaithersberg, MD) instead of goat anti-human IgG (H&L) (KPL Products)24. The specificity of the anti-IgM reagent had previously been shown to be specific for m-chain antibodies and did not react with IgG antibodies15, 24.
Point 2: a) How IgM are specific?
Response 2a: The specificity of the anti-IgM reagent for μ chain specificity was tested previously in two ways: 1) with serum from HIV+ patients with heavy staining IgM antibody bands to P. jirovecii that had been separated by column chromatography into IgM and IgG fractions; and 2) with purified IgM and IgG on immunoblots15. In both cases, anti-IgM antibody only reacted with the IgM preparations, indicating that it was μ chain-specific. As an additional measure of specificity, 200 healthy blood donors were also screened for IgM and IgG using the same IgG and IgM ELISA. Of 200 serum samples examined, IgG antibodies were detected in 103 (51.5%) of the specimens, whereas IgM antibodies were only detected in 7 (3.5%) of the specimens (p<0.001)24.
- b) a specific IgM can stay for long in blood:
Response 2b: We have not evaluated whether anti P. jirovecii-specific IgM can stay in the blood for long. However, to test this needs follow up of individuals post a known episode of P. jirovecii colonization. This is an interesting hypothesis that can be explored further. We have indicated that the infections in adults or children tested were asymptomatic. None of the individuals participating in the study were suspected of previously or currently having Pneumocystis pneumonia. This is now indicated in the description of the patients in the methods section and also, in the results.
Point 3:: How IgG are specific?
Response 3: Please refer to the answer to Points 1 and 2a.
Point 4: The different specificity of IgM and IgG can explain the results obtained.
Response 4: Please refer to the answer of Point 2a: Of 200 serum samples examined, IgG antibodies were detected in 103 (51.5%) of the specimens, whereas IgM antibodies were only detected in 7 (3.5%) of the specimens (p<0.001)24.
